# Characterizing the Pathogenesis and Immune Response of Equine Herpesvirus 8 Infection in Lung of Mice

**DOI:** 10.3390/ani12192495

**Published:** 2022-09-20

**Authors:** Leyu Hu, Tongtong Wang, Huiying Ren, Wenqiang Liu, Yubao Li, Changfa Wang, Liangliang Li

**Affiliations:** 1College of Agronomy, Liaocheng University, Liaocheng 252000, China; 2College of Veterinary Medicine, Qingdao Agricultural University, Qingdao 266109, China

**Keywords:** EHV-8, BALB/c mice, respiratory diseases, pathogenicity, proinflammatory cytokines

## Abstract

**Simple Summary:**

Equine herpesvirus 8 (EHV-8) is an important pathogen primarily affecting the horse and donkey industry, but there is little information about the pathogenicity and immune response of EHV-8 in a mouse model. We aim to investigate the pathogenicity and immune response in the lung during EHV-8 infection in BALB/c mice. The results showed that EHV-8 could effectively replicate and elicits a strong proinflammatory response in the lung tissues of a mouse model. The mouse model of viral respiratory disease proposed here will also be useful for studying the underlying mechanisms of the pathology of respiration.

**Abstract:**

Equine herpesvirus type 8 (EHV-8), associated with abortion and severe respiratory disease in donkeys and horses, causes significant economic losses in the global equine industry. However, the pathogenicity of EHV-8 is still unknown. Mice are widely used as an animal model to evaluate virus replication and virulence. The present study aimed to evaluate the pathogenicity of the EHV-8 SDLC66 strain in BALB/c mice. Mice were used to test for infection-associated parameters (such as clinical signs, body weights, virus replication in tissues, viremia, and cytokines) and sacrificed at 0, 2, 4, and 6 days post-infection (dpi). The mice inoculated with EHV-8 exhibited lethargy, dyspnea signs, loss in body weight, and viremia. EHV-8 was detected in the liver, spleen, brain, and lung by PCR at 4 dpi and 6 dpi, effectively replicating these tissues detected by TCID_50_ at 6 dpi. Proinflammatory cytokines, including IL-6, IL-1β, and TNF-α, were significantly increased at the 4 dpi and 6 dpi in the lung than in the control group. However, IFN-γ was only increased at 6 dpi in the EHV-8-infected group. These data showed that EHV-8 could enter the lungs of mice and cause respiratory disease in the mouse model, which helps reveal the pathogenicity of EHV-8.

## 1. Introduction

The equine herpesviruses are infectious pathogens that cause serious respiratory disease and abortion in the equine or donkey industry. Until now, nine herpesviruses have been identified in equids. Horses were considered the natural host to EHV-1~5, while donkeys are the host to EHV-6~8 (AHV-1~3), Thomson’s gazelle, giraffe, and polar bear are the natural host of EHV-9 [1,2,3]. EHV-8, along with EHV-1, EHV-3, EHV-4, and EHV-9, belongs to the subfamily *Alphaherpesvirinae* [4,5], characterized by short replication cycles, destruction of the host cell [6,7,8]. 

EHV-8 is a double-stranded enveloped DNA virus of 150 kb in length, which contains at least 76 open reading frames (ORFs). Numerous EHV-8 strains have been isolated and characterized in various countries. For instance, EHV-8 was first reported in Australia and isolated from a donkey in 1987 [9], it was also reported from horses in China in 2010, named EHV-8 wh strain [6], and another report showed that EHV-8 was isolated from donkeys in Israel in 2020 [10]. The development of large-scale donkey breeding has been rapid in China in recent years. However, abortion and respiratory disease seriously hinder the healthy development of the donkey industry. Our group found that abortion and neurological disease cases in donkeys are closely related to EHV-8 infection [11,12]. More notably, the seroprevalence rates of EHV-8 are high in Shandong Province, reaching 31% [12]. These data highlight the potential threat of EHV-8 to the donkey industry. However, the pathogenicity and immune responses of EHV-8 infection in the lung are still unknown. 

Mouse models have recently been used to evaluate the pathogenicity and immune response in EHV-1 and EHV-9 [13,14,15,16,17]. The correlational research of EHV-8 in mouse models was limited. In the present study, we aimed to evaluate the pathogenicity and cytokine responses in lung tissue of BALB/c mice.

## 2. Materials and Methods

### 2.1. Virus Culture 

The rabbit kidney 13 (RK-13) cells were cultured by 10% FBS minimum essential medium (MEM) at 37 °C in a humidified atmosphere of 5% CO_2_ and used for propagation of the EHV-8 SDLC66 strain (GenBank accession: MW816102). The virus was titrated by a plaque formation assay as previously described [18], and the viral titer in the suspension used for inoculation was 1×10^6^ plaque-forming units (PFUs)/mL. 

### 2.2. Animal Design and Ethics Statement

Twenty-four specific pathogen-free, 8-week-old male BALB/c mice, were obtained from the Experimental Animal Center of Shandong University and tested using both RT-PCR/PCR to confirm that they were EHV-8, EHV-1, EHV-4, and EAV negative. After 1 week of adaptation, the mice were randomly allocated to two groups (*n* = 12 mice/group), and each group was housed separately to prevent cross-infection. Group 1 was inoculated intranasally with 100 µL MEM and served as the control group. Group 2 was inoculated intranasally with 100 µL EHV-8 solution (1 × 10^5^ PFU/mice); EHV-8 or MEM incubation in mice was performed under deep anesthesia with Zoletil 50 (Virbac, Nice, France). The mice were observed daily for any clinical signs and euthanized at 0, 2, 4, and 6 days post-infection (dpi), and a careful postmortem examination was performed immediately for sample collection. 

### 2.3. Clinical Evaluation and Sample Collection

All mice were assessed clinically for clinical signs and body weight loss. Meanwhile, the tissue samples (brain, lung, liver, spleen, and kidney) and serum were collected to test EHV-8 replication and viremia. In addition, the lungs were fixed with 10% formalin for further histopathological and immunohistochemical analysis and used to detect cytokine transcript.

### 2.4. Virus Replicates in Tissues

Viral genomic DNA was extracted from serum or tissues using a Viral DNA Kit (Omega Bio-Tek, Inc., Norcross, GA, USA) according to the manufacturer’s protocol. To determine the EHV-8 replication in mice, a real-time PCR assay was used to detect EHV-8 in serum based on part ORF70 gene amplification with G1 F and R primers (Table 1). The amplicons were inserted into the pMD18-T vector to generate recombinant plasmids, which were used to develop the real-time PCR assay as previously described [19]. Simultaneously, PCR was performed with G2 F and R primers (Table 1) to detect the EHV-8 distribution in mice tissues at 2, 4, and 6 dpi. 

The virus titers in tissues of EHV-8-infected mice were further determined in RK-13 cells. Briefly, three mice from group 2 were sacrificed at 6 dpi, and their liver, spleen, lung, kidney, and brain were collected. These tissues (0.1 g) mixed with PBS (1 mL) were crushed, homogenized, then frozen and thawed 3 times. After that, the supernatant was collected from different tissues, filtered through a 0.22 µm syringe filter, and tittered in RK-13 cells using the Reed–Muench method described previously [12].

### 2.5. Histopathology and Immunohistochemistry Evaluation 

The mice lungs of the control group and EHV-8 infected group at 6 dpi were collected and fixed with 10% formalin, embedded in paraffin wax, sliced in a microtome (Leica, Nussloch, Germany) to 4 µm, affixed onto slides, and then subjected to hematoxylin and eosin (HE) staining for microscopic examination and immunohistochemistry (IHC) staining for detection of EHV-8 antigen with positive serum as described previously [12]. Briefly, slides were boiled with sodium citrate (pH 6.0) for antigen retrieval, then treated with hydrogen peroxide to inhibit endogenous peroxidase. After blocking with 5% BSA, slides were incubated with EHV-8-positive serum overnight at 4 °C; thereafter, the slides were rinsed in PBS, treated with horseradish peroxidase-conjugated goat anti-mouse IgG for 1 h at 37 °C, rinsed in PBS again, and stained with diaminobenzidine for 5 min, then rinsed in water to stop the reaction. Finally, slides were counterstained in Gill’s hematoxylin for 30 s, dehydrated, cleared, and placed on a coverslip to be observed by light microscopy. Negative controls were treated without antibody incubation to evaluate non-specific binding effects. 

### 2.6. Quantification of Cytokines mRNA

The total RNA of lungs from the different group were extracted using TRIzol (Takara, Osaka, Japan), these RNA concentrations were quantified by spectrophotometer Biophotometer B500 (METASH, Shanghai, China), and 1 µg of RNA was reversed transcription into cDNA with the One-Step RT-PCR Kit (Takara, Osaka, Japan). The reaction system (final volume of 30 µL) included 15 µL of 2× RealStar Green Fast Mixture (GenStar, Beijing, China), 2 µL cDNA, 2 µL corresponding primers, and 11 µL of DEPC water. Quantitative PCR (qPCR) analyses were performed on a CFX Connect real-time PCR system (Bio-Rad) using SYBR Premix Ex Taq II DNA polymerase (Takara, Osaka, Japan). The following thermocycling conditions were used for qPCR: initial denaturation at 95 °C for 5 min; 40 cycles at 95 °C for 10 s; 60 °C for 30 s; 72 °C for 20 s. The expressions of IL-6, IL-1β, IFN-γ, and TNF-α were normalized against those of GADPH by the 2^−ΔΔCT^ threshold cycle (CT) method. All test samples were run in three independent experiments, and these primers are presented in Table 1.

### 2.7. Statistical Analysis

Data are expressed as the mean ± standard error of the mean (SEM) or the mean ± standard deviation (SD). Differences among groups were analyzed by one-way analysis of variance followed by the Bonferroni post hoc test or unpaired *t*-tests. *p*-values of < 0.05 or less were considered significant.

## 3. Results

### 3.1. Clinical and Necropsy Findings

Clinical signs of the EHV-8 infection group, including lethargy, ruffled coat, dyspnea, and crouching in corners, and neurological signs in some mice were observed from 4 dpi (Figure 1A). As expected, there were no obvious clinical signs observed in the MEM-inoculated mice at any time throughout the experiment (Figure 1B). Compared with the normal group (Figure 1C), hemorrhage lesions were found in EHV-8 infected group at 6 dpi (Figure 1D). 

### 3.2. EHV-8 Replicates Efficiently in BALB/c Mice

To determine EHV-8 replication in mice, viral DNA was extracted from serum, and EHV-8 was detected from 2 to 6 dpi with qPCR, with the peak titer at 6 dpi. No EHV-8 was detected in the sera of the MEM-infected mice throughout the experiment (Figure 2A). Furthermore, EHV-8 DNA detection from several organs of BALB/c mice was performed by PCR with specific primers. As shown in Table 2, it was detected in the lung, brain, and other tissues (liver and spleen) at 4 and 6 dpi. However, the kidney was negative. 

Further, EHV-8 in the tissues of infected mice was titrated on RK-13 cells. The mean titers are 1.6 × 10^4^ TCID_50_ in the lung, 4 × 10^3^ TCID_50_ in the brain, 6.25 TCID_50_ in the liver, and 16 TCID_50_ in the spleen, respectively (Figure 2B). During the observation period, the mice had lost up to 34% mean body weight by 6 dpi (Figure 2C), and EHV-8 infected mice showed significant mortality (2/3) compared to control group mice at 6 dpi (Figure 2D). These data showed that the SDLC66 replicated efficiently in mice.

### 3.3. Histopathological Evaluation and Immunohistochemical Detection of Viral Antigen

EHV-8 infection can cause respiratory symptoms in the equines. To explore the pathogenesis of EHV-8 in vivo, we tested pathological lesions of lung tissues and compared them between the infected group and the control group. The results showed that EHV-8 infection induced severe histological lesions in the lung tissue. EHV-8-infected mice showed severe interstitial pneumonia (characterized by thicker alveolus walls, inflammatory cell infiltration, and hemorrhage) (Figure 3A). Simultaneously, EHV-8 was detected in the lungs of the mice infected with EHV-8 by immunohistochemistry using anti-EHV-8 antibodies (Figure 3B). 

### 3.4. Cytokine Expression Patterns after EHV-8 Infection

To examine IL-6, IL-1β, IFN-γ, and TNF-α mRNA expression in mice lungs following infection with EHV-8, we inoculated BALB/c mice with EHV-8 and analyzed them by qPCR. The result showed that all IL-6, IL-1β, and TNF-α mRNA were significantly increased at 4 dpi and 6 dpi in the lung. However, IFN-γ mRNA was significantly increased at 6 dpi (Figure 4).

## 4. Discussion

EHV-8 infection is closely associated with abortion, respiratory symptoms, and neurological illness in equines [1,11,12]. The mouse model is suitable for viral pathogen studies due to its low cost, easy manipulation, and well-characterized genetic background [20]. We also investigated the host’s viral behavior and immune response [21,22]. Several reports have indicated that mice can serve as animal models for EHVs infection [12,19,23,24]. However, the pathogenicity and immune response of EHV-8 in mice is still unknown. This study detected the viremia, tissue-specific distribution of EHV-8, and proinflammatory cytokines in BALB/c mice with systematic analysis. These data indicate that EHV-8 can replicate effectively in the mouse model.

In the current study, the pathological findings of EHV-8 were interstitial pneumonia in the lung of BALB/c mice, which was similar to results observed for EHV-1 infection in a previous study [25,26,27]. However, the infection of EHV-9 in BALB/c mice develops necrosis of the olfactory epithelium [24]. Notably, the EHV-8 ORF70 gene was detected in the brain, which was closely related to viral encephalitis in mice in a previous study [12]. Meanwhile, the ORF70 gene was also detected in the liver and spleen, and our study offers new insights into the pathogenicity of EHV-8 in mice. 

In response to viral infection, the expression of proinflammatory cytokines and chemokines is upregulated at both transcriptional and translational levels [28,29]. Proinflammatory cytokines, such as IL-1α, IL-1β, IL-6, IL-12β, and TNF, were upregulated within the brain of EHV-1-infected mice [16]. The proinflammatory cytokines IL-15, IL-6, IFN-γ, MIP-1a, MIP-1b, MIP-2, and TNF-α were commonly expressed in lung tissue during EHV-1 infections in the previous study [30]. Another research demonstrated increased IFN-γ and IL-4 responses post-challenge by EHV-1 in ponies [31]. However, the cytokine’s response to EHV-8 infection in the lung is not fully understood. In the present study, for EHV-8 infection in mice, the expression of cytokines IL-6, IL-1β, and TNF-α was increased at 4 dpi and 6 dpi in the lung, which might be closely related associated with tissue damage.

## 5. Conclusions

In short, BALB/c mice are susceptible to EHV-8 infection, as evidenced by viremia and lung damage. These data suggest that the mouse model could be a substitution in studying future host–pathogen interactions and vaccine candidates for EHV-8 in vivo.

## Figures and Tables

**Figure 1 animals-12-02495-f001:**
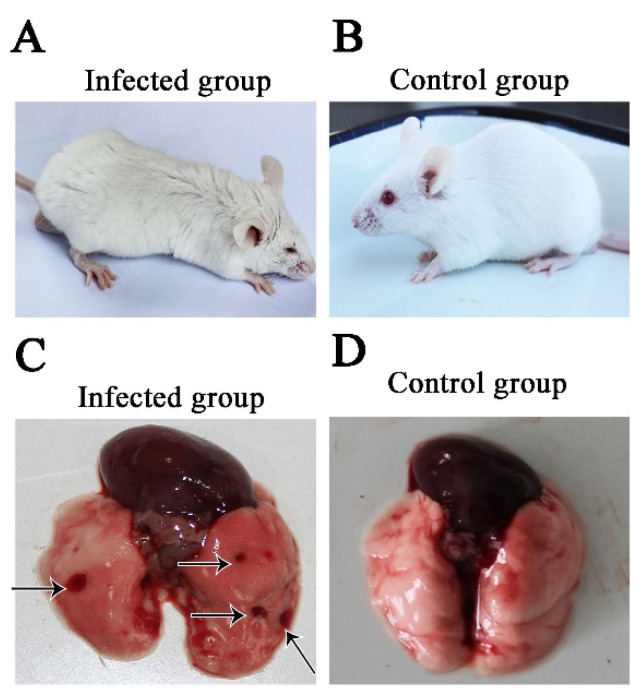
**Clinical and Necropsy Findings**. Twenty-four specific pathogen-free mice were randomly allocated to two groups (group 1 was a control, and group 2 was inoculated intranasally with EHV-8). Clinical signs of the EHV-8 infected group (**A**) and control group (**B**) were monitored for 7 days. The pathological changes in mice lungs were observed by necropsy findings at 6 dpi. (**C**) Hemorrhage lesions (Arrows indicated) in the lung of the EHV-8 infected group, and (**D**) normal lung lesions of the control group.

**Figure 2 animals-12-02495-f002:**
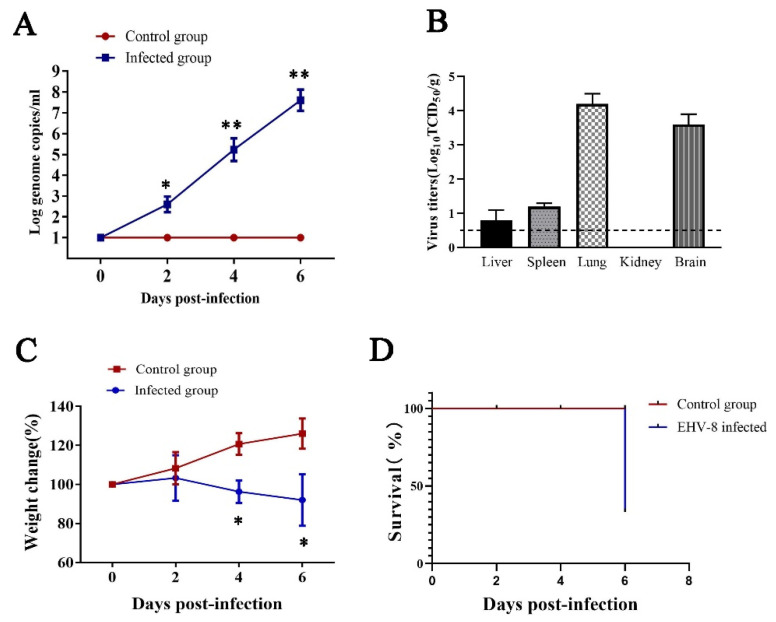
**The EHV-8 replicates efficiently in BALB/c mice.** Viremia was detected in serum at the indicated time points (**A**). EHV-8 replication in tissues at 6 dpi was also tested by titrating on RK-13 cells (**B**), body weight (**C**), and mortality (**D**). The data shown are representatives from three independent experiments and subjected to Student’s *t*-test. * *p* < 0.05 vs. the control group mice at the same time point; ** *p* < 0.01 vs. the control group mice at the same time point.

**Figure 3 animals-12-02495-f003:**
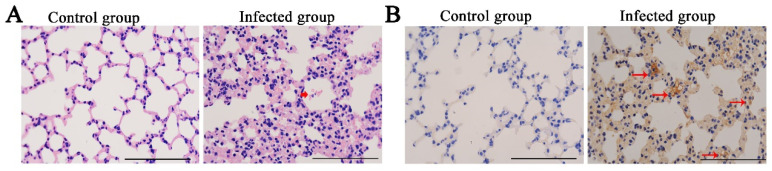
**Histopathology and Immunohistochemistry Evaluation**. The lungs from both group 1 and group 2 mice were fixed at 6 dpi to perform a histopathological examination by H.E (**A**) and immunohistochemistry (IHC) for EHV-8 antigen detection using EHV-8-positive serum (**B**). Bar: 100 μm. Broad arrow indicated hemorrhage; narrow arrow indicated EHV-8.

**Figure 4 animals-12-02495-f004:**
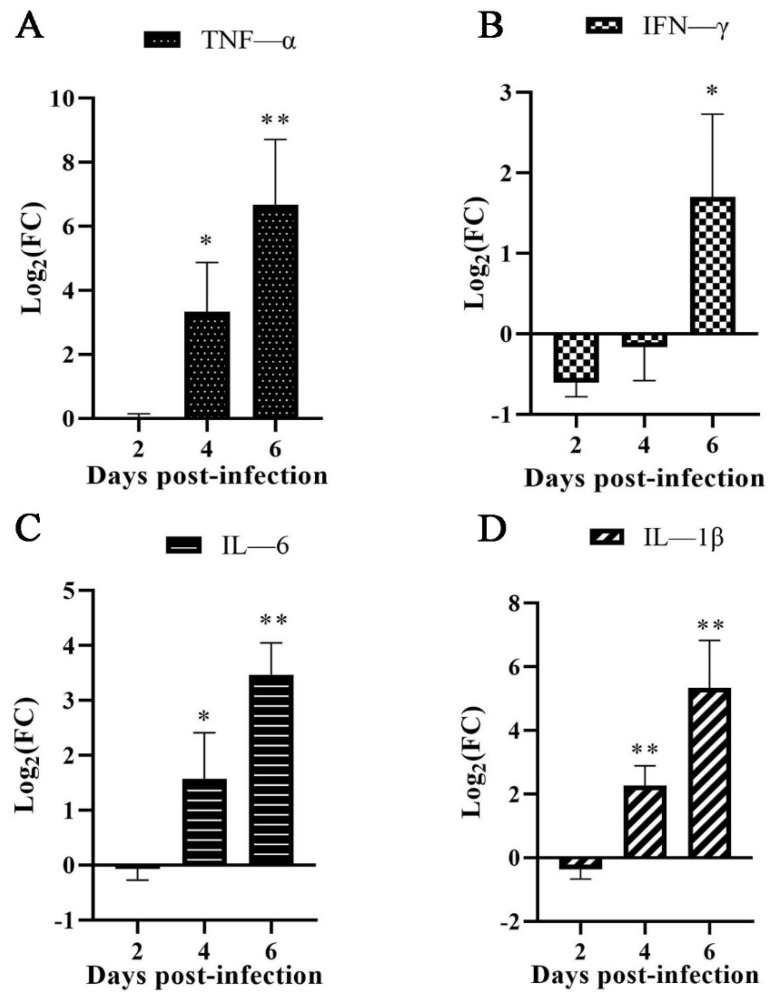
**Quantitative PCR (qPCR) detection transcript of cytokines in the lung**. Total RNA was extracted from the control group and EHV-8 infected group, and cDNA was generated using the One-Step RT-PCR Kit. The relative gene expression of TNF-α (**A**), IFN-γ (**B**), IL-6 (**C**), and IL-1β (**D**) cytokines in the EHV-8 infected group compared to a control group was assessed using qPCR assays in samples of lungs from mice. Data are expressed as fold change (FC) estimates (log2(FC)). Asterisks indicate statistical significance, * *p* < 0.05; ** *p* < 0.01. Primer melting curve can be found in Appendix A.

**Table 1 animals-12-02495-t001:** Primers for EHV-8 detection or cytokines expression.

Genes	Forward Primer (5’-3’)	Reverse Primer (5’-3’)
EHV-8-G1	ACTCCAGTGCAGCGGATTCGTC	GTCCAATGAGAGCCAAGCAAAT
EHV-8-G2	TCAGACTGTCACTCGTGGGA	CCTGAAGGCCGTTTAACACA
IFN-γ	GCTCTGAGACAATGAACGCTAC	TCTTCCACATCTATGCCACT
TNF-α	ACGGCATGGATCTCAAAGAC	GTGGGTGAGGAGCACGTAGT
IL-6	GCTGCTTCCAAACCTTTGAC	AGCTTCTCCACAGCCACAAT
IL-1β	CCGGAGAGGAGACTTCACAG	CAGAATTGCCATTGCACAAC
GAPDH	CCCCTGGCCAAGGTCATCCATG	GGCCATGAGGTCCACCACCCTGT

**Table 2 animals-12-02495-t002:** EHV-8 detection in several organs of BALB/c mice by PCR.

dpi	Lung	Brain	Spleen	Liver	Kidney
2	0/3	0/3	0/3	0/3	0/3
4	3/3	3/3	2/3	1/3	0/3
6	3/3	3/3	2/3	1/3	0/3

## Data Availability

The data that support the findings of this study are available from the authors.

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
