# Peer review of "Characterizing the Pathogenesis and Immune Response of Equine Herpesvirus 8 Infection in Lung of Mice"

_animals, 2022, doi:10.3390/ani12192495_

Round 1

Author Response

Abstract:

The first sentence of the abstract is problematic "The equine herpesvirus type 8 (EHV-8) causes significant economic losses in the global horses and donkey industry, which has been associated with abortion and severe respiratory disease in the equine." It should be reworded as: "Equine herpesvirus type 8 (EHV-8), which has been associated with abortion and severe respiratory disease in donkeys and horses, causes significant economic losses in the global equine industry."

Answer: We thank you for the comments. The sentence has been changed to "Equine herpesvirus type 8 (EHV-8), associated with abortion and severe respiratory disease in donkeys and horses, causes significant economic losses in the global equine industry", as your suggestion in lines 21-23 marked red in the revised manuscript.

Line 10: "As is known to us" is completely unnecessary.

Answer: We have deleted the sentence as suggested in the revised manuscript.

Line 16: Avoid using the idiom "could be" when stating results. It implies uncertainty. The last sentence is a declaration that you will need to back up later. How does it help reveal the pathogenesis of EHV-8? You have too many of your methods and results in the abstract. Why would I want to read the paper when I can get all the information from the abstract?

Answer: Thanks for your comments. We have changed "could be" to "was" in lines 28, 136, and 169, marked in red in the revised manuscript, and re-edited the abstract section in the revised manuscript.

Introduction:

Introduction is way too brief. It is only two paragraphs! Expound more on the virus, its history, whether it only affects domesticated equines or is found in wild populations, its zoonotic potential how is it related to other equine herpesviruses, etc.

Answer: We thank you for your suggestion. The introduction section was re-edited with more information about EHV-8, shown in lines 39-57, marked red in the revised manuscript.

Line 29: You are including EHV-8 in a group of other closely related EHV's, so "they" belong to the subfamily Alphaherpesvirus. End the sentence there. The second half of the sentence makes little sense in relation to the first. Are you describing EHV-8 or all the others in that subfamily?

Answer: Thanks for your suggestion. We have re-edited these sentences marked in red in lines 39-41 in the revised manuscript.

Line 36: "The above studies demonstrate that EHV-8 infection is closely related to respiratory disease, abortion, and neurological symptoms in the equine." EHV-8 is closely related to respiratory disease and abortion in general? This sentence also makes little sense.

Answer: Thanks for your suggestion. We re-edited this sentence in lines 48-49, marked red in the revised manuscript.

Line 38: mouse models not mice models

Answer: We appreciate your suggestion. We have changed the words in lines 14, 17, 33, 53, 54, and 183 marked red in the revised manuscript.

Line 39: expand on why the correlational research was limited. Is this the first mouse study with EHV-8? Why did you chose a mouse model? Why did you chose this particular strain of mouse? Where are your aims and hypotheses?

Answer: Thanks for your comment. We reply to these questions point to point:

  1. There have been few studies on EHV-8 until now, so our group chooses the BALB/c mice to test EHV-8 infection for the first time.
  2. The mouse model is suitable for viral pathogen studies due to its low cost, easy manipulation, and well-characterized genetic background.
  3. EHV-8 could replicate efficiently in BALB/c mice, tested in our group's previous study, "The emergence of viral encephalitis in donkey by equid herpesvirus 8 in China" (Wang et al., 2022).
  4. We added the aims and hypotheses in the Simple Summary section on lines 13-19, marked red in the revised manuscript.

Methods:

You state that the titer you grew was 1 x 10^6 PFU/ml. but the titer you gave to each mouse was 1 x10^5 because you only gave them 100 µl. You need to be more clear about this. Were the mice anesthetized when you infected them? If so, name the anesthetic .Did you euthanize mice at day 0?

Answer: Thanks for your suggestion. We have added more information in this part:

  1. Group 2 was inoculated intranasally with 100 µl EHV-8 solution (1×105 PFU/mice). EHV-8 or MEM incubation in mice was performed under deep anesthesia with Zoletil 50 (Virbac, France), marked red on lines 70-72 in the revised manuscript.
  2. Yes, we euthanized mice at day 0, shown in lines 72 marked red in the revised manuscript.

Viral replicates section, You should have separate sections for RNA titers and infectious viral titers. Describe your TCID50 methods!

Answer: We appreciate your suggestion. We re-edited the viral replicates section: "The virus titers in tissues of EHV-8 infected mice were further determined in RK-13 cells. Briefly, three mice from group 2 were sacrificed on 6 dpi, and their liver, spleen, lung, kidney, and brain were collected. These tissues (0.1 g) mixed with PBS (1 mL) were crushed, homogenized, then frozen and thawed 3 times. After that, the supernatant was collected from different tissues, filtered through a 0.22-µm syringe filter, and tittered in RK-13 cells using the Reed–Muench method described previously", marked red in lines 88-93 in the revised manuscript.

Your qPCR methods are insufficiently described

Answer: Thanks for your comments. The qPCR section was re-edited with more detail marked red in lines 108-118 in the revised manuscript.

Did you perfuse the lungs with fixative before you collected them? IHC methods are not well described, no mention of blocking, antigen retrieval, Ab used, etc.

Answer: Thanks for your comments and suggestion. The lungs were collected from a different group and cut into 2 parts, finally immersed in 10% formalin to fix. Meanwhile, the IHC methods were re-edited with more details marked red in lines 95-106 in the revised manuscript.

Results:

Clinical signs: neurological? Describe this in more detail.

Figure 1: Should be A, B, C and D because you have 4 pictures. They should all be the same size, as well. Remove the titles above the picture and describe what each picture is in the caption.

Answer: Thanks for your comments and suggestion. We have re-edited the clinical signs, and Figure1, including the figure legend, marked red in lines 126-130 and 302-305 in the revised manuscript.

3.2. Remove "The" from title

Line 114: RNA is detected in the organs not from You extracted viral RNA not DNA correct? Your TCID50 results are infectious viral titers and your RT-PCR results are just viral titers. TCID50 is more important and should be in separate sections of results.

Answer: Thanks for your comments. We have removed "The" in the 3.2 titles. Viral DNA is correct. TCID50 sections have been separated in results.

Fig 2B: Y axis is missing units Log10TCID50/g of tissue. Did you weigh and record the tissues you collected at necropsy? That is part of the calculations when titering RNA or infectious for tissues.

Answer: As suggested, we re-edited the Y-axis units with Log10TCID50/g of Fig 2B. The tissues of mice were weighed and recorded during the collection at necropsy.   Thank you.

Fig 2C: A better calculus would be % of weight loss from the original weight before infection. Also, Fig 2, x axis label should be Days, post-infection.

Answer: We have changed the calculation method in weight indicated % of weight loss, and the x-axis label should be changed to Days post-infection.

Table 2 title: detection "IN" several organs also you should have graphed your titer results for the organs. Table is not well explained, and your numbers don't match up because you said in Fig 2 that you did 3 independent experiments without being specific as to what you did. Did you do three mouse experiments?

Answer: Thanks for your comments. We changed the title of Table 2, marked red in the revised manuscript. We detected EHV-8 antigen in different organs (Lung, Brain, Spleen, Liver, and Kidney) of mice by PCR at 2, 4, and 6 dpi. The number in Table 2 represents the number of mice was EHV-8 positive.

Figure 4: You should not have your controls on qPCR graph. Your gene expression results are fold-change differences compared to the levels in your mock mice, which should be around 1. If you did double delta CT correctly you would subtract your infected results from those of your control mice and that would be your fold change increase or decrease. What house keeping gene or gene did you use? Where is the data for your primer efficiency?

Answer: We have revised Figure 4 by comparing cytokine transcript levels between the infected and control groups; 4 cytokines were separated, and the housekeeping gene is GAPDH. We also provide the melting curve, which was indicated in supplemental materials.

  1. Discussion:

Insufficient-mostly repeating your results.

Answer: Thanks for your comments and suggestion. The EHV-9 pathology in line 168-169 marked red in the revised manuscript was added in the discussion section.

Reviewer 2 Report

This is a well-planned research. It is very concise, clearly written and easy to understand. Hu and colleagues studied the pathogenesis of Equine Herpesvirus type 8 strain SDLC66 in BALB/c mice. According to the authors, all the animal testing were done under ethical conditions. They found BALB/c mice to be susceptible to EHV-8 infection.  The virus replicated very well especially in the lung and brain tissues, but also in the liver and spleen tissues. They did not detect any virus transcripts in the kidney. EHV-8 infection induced distinct histological lesions in the lungs of mice which were typical of severe interstitial pneumonia. Hu and colleagues also observed that the EHV-8 infection induced brain lesions in the mice similar to the ones they had previously seen in the brain tissues of mice and donkeys in their March, 2022 published research article. These brain lesions were characteristic of viral encephalitis. The observations from this clinical studies using mice give a credible explanation for the respiratory and neurological disease symptoms observed in EHV-8 infected horses and donkeys. I recommend that this manuscript is published. 

Author Response

Thank you for your suggestion. We have re-edited the language of manuscript.

Reviewer 3 Report

In the present manuscript, the author reported the pathogenesis of the EHV-8 SDLC66 strain in BALB/c mice. EHV-8 infection can induce interstitial pneumonia and increased expression of cytokines, which might be closely related to tissue damage. It is undoubtedly a meaningful finding that helps reveal the pathogenesis of EHV-8 in equines. However, there are a few points that need the author to improve and address:

1.       In lines 41-42, sentences show that “The above studies demonstrate that EHV-8 infection is closely related to respiratory disease, abortion, and neurological symptoms in the equine”, the histopathology and immunohistochemistry evaluation only have the data on lung, not brain or reproductive system, why?

2.       In line 61, “dpi” should be spelled full name for the first time and abbreviation in line 100.

3.       In line 34, “~8(AHV-1~3)” should be changed to “~8 (AHV-1~3)”.

4.       In Figure 1B, hemorrhage in the lung should be pointed out with an arrow.

5.       In lines 233-234, the sentence “group 1 was control. and group 2 was inoculated intranasally with EHV-8 solution for 0, 2, 4, and 6 days” should be changed to “group 1 was a control, and group 2 was inoculated intranasally with EHV-8 solution for 0, 2, 4, and 6 days”.

Author Response

  1. In lines 41-42, sentences show that "The above studies demonstrate that EHV-8 infection is closely related to respiratory disease, abortion, and neurological symptoms in the equine", the histopathology and immunohistochemistry evaluation only have the data on lung, not brain or reproductive system, why?

Answer: Thanks for your comments. We performed histopathology and immunohistochemistry evaluation in the brain from the mouse model. These data have been published in our previous study, "The emergence of viral encephalitis in donkey by equid herpesvirus 8 in China", and we are performing the pathogenicity in reproductive system study with a female mouse model.

  1. In line 61, "dpi" should be spelled full name for the first time and abbreviation in line 100.

Answer: We have re-edited days post-infection (dpi) according to your suggestion in lines 73 and 127 in revised manuscripts, Thanks!

  1. In line 34, "~8(AHV-1~3)" should be changed to "~8 (AHV-1~3)".

Answer: Thanks for your comments. According to your suggestion, we have added a space in line 39 in the revised manuscript.

  1. In Figure 1B, hemorrhage in the lung should be pointed out with an arrow.

Answer: Thanks for your suggestion. We have added an arrow in revised Figure 1C in the revised manuscript.

  1. In lines 233-234, the sentence "group 1 was control. and group 2 was inoculated intranasally with EHV-8 solution for 0, 2, 4, and 6 days" should be changed to "group 1 was a control, and group 2 was inoculated intranasally with EHV-8 solution for 0, 2, 4, and 6 days".

Answer: Thanks for your comments. We have re-edited the sentence in lines 301-302, marked red in revised manuscripts.

Reviewer 4 Report

In this manuscript author used mice as animal model to evaluate EHV-8 virus replication and virulence. After 4 days the mice inoculated with virus showed clinical signs, meanwhile viruses were also detected in the serum and many tissues. The lung and brain contained the higher amount of virus than in other tissues by TCID50. Apart that, author also found some specific cytokines were largely increased in lung of infected mice than in the control group. These data sufficiently approved that mice could be animal model to investigate the pathogenesis of EHV-8.

Whereas there are still some minors to improve:

In general, ‘Figure with numbers,1, 2…..’ doesn’t need to be on the top of  individual figure additionally, as which has stated in figure legend.

Line 29: 9 herpesviruses were identified in equids, author should state the host of EHV-9.

Line 66: EHV-8 also caused the neurological signs, plus the amount of Virus RNA and virus titers were higher in both lung and brain, but why here author only did histopathological and immunohistochemical analysis on lung?

Line 81: lungs from both group 1 and group 2 mice were fixed, right? Not only group 2.

Line 113: ‘MEM-infected mice’, which is not ‘sham-infected mice’, editing error.

Line 117: Could author change the format of TCID50? For instance, 1,6x104  instead of 104.2 TCID50 etc. If this way give a better estimation about the amount of virus.

Figure 2:  Could author present infected group and control group in A and C in the same color? Or it is better that choose other colors other than deep blue for either infected group or control group, as it is hard to distinguish between black and deep blue.

Figure 4: Which cytokine was detected in control group? If author intended to compare cytokines transcript level between infected and control group, should it be in parallel with all 4 cytokines? instead of only one column. In addition, detecting cytokines transcript by Q-PCR might not perfectly reflect the actual level cytokines secreted, has or can author try to detect cytokines via ELISA kit or cell flow cytometry?

Author Response

In general, 'Figure with numbers,1, 2…..' doesn't need to be on the top of  individual figure additionally, as which has stated in figure legend.

Answer: Thanks for your comments. According to your suggestion, we have removed the Figure numbers from the pictures in the revised manuscript.

Line 29: 9 herpesviruses were identified in equids, author should state the host of EHV-9.

Answer: Thanks for your comments and suggestion. We have added the information on the host of EHV-9 "Thomson's gazelle, giraffe, and polar bear are the natural host of EHV-9 (Hideto Fukushi et al., 2012)", which was marked in red in line 39 in the revised manuscript.

Line 66: EHV-8 also caused the neurological signs, plus the amount of Virus RNA and virus titers were higher in both lung and brain, but why here author only did histopathological and immunohistochemical analysis on lung?

Answer: Thanks for your suggestion. We performed histopathology and immunohistochemistry evaluation in the brain from the mouse model in the previous study. This data has been published in Front Microbiol with the title "The emergence of viral encephalitis in donkey by equid herpesvirus 8 in China".

Line 81: lungs from group 1 and group 2 mice were fixed, right? Not only group 2.

Answer: Yes, we have re-edited the description in Figure legends of Figure 3 in the revised manuscript, which were marked in red. Thanks.

Line 113: 'MEM-infected mice', which is not 'sham-infected mice', editing error.

Answer: Thanks for your suggestion. We have changed the sentence with MEM-infected mice, marked in red in line 134 in the revised manuscript.

Line 117: Could author change the format of TCID50? For instance, 1,6x104  instead of 104.2 TCID50 etc. If this way give a better estimation about the amount of virus.

Answer: Thanks for your comments. We have changed the format of TCID50 in line 139 in the revised manuscript. Thanks.

Figure 2:  Could author present infected group and control group in A and C in the same color? Or it is better that choose other colors other than deep blue for either infected group or control group, as it is hard to distinguish between black and deep blue.

Answer: Thanks for your suggestion. We have re-edited Figure 2A and 2C per your suggestion in the revised manuscript.

Figure 4: Which cytokine was detected in control group? If author intended to compare cytokines transcript level between infected and control group, should it be in parallel with all 4 cytokines? instead of only one column. In addition, detecting cytokines transcript by Q-PCR might not perfectly reflect the actual level cytokines secreted, has or can author try to detect cytokines via ELISA kit or cell flow cytometry?

Answer: Thanks for your comments. The expression of IL-6, IL-1β, IFN-γ, and TNF-α in mouse lungs was detected in the control group. We have re-edited the description in the figure legend of Figure 4, and readjusted Figure 4.

Round 2

Reviewer 1 Report

The controls should be removed from the qPCR graphs.  The relative expression in experimental mice is: Expression level of experimental mice minus expression level of control mice (controls or mocks should be around 1), thus the control levels should not be on the graph.

Author Response

The controls should be removed from the qPCR graphs.  The relative expression in experimental mice is: Expression level of experimental mice minus expression level of control mice (controls or mocks should be around 1), thus the control levels should not be on the graph.

Answer: We have removed the control in qPCR graphs, which indicated “Expression level of experimental mice minus expression level of control mice” in the revised Figure4. The Figure 4 legend was re-edited to “Fig. 4. Quantitative PCR (qPCR) detection of cytokines transcript in the lung. Total RNA was extracted from the control group and EHV-8 infected group, and cDNA was generated using the One-Step RT-PCR Kit. The relative gene expression of TNF-α (A), IFN-γ (B), IL-6 (C), and IL-1β (D) cytokines in the EHV-8 infected group compared to a control group was assessed using qPCR assays in lung samples. Data are expressed as fold change (FC) estimates [log2(FC)]. Asterisks indicate statistical significance, *P <0.05; **P <0.01” in lines 321-331 in the revised manuscript. Thanks!
